# Sex Hormone-Binding Globulin (SHBG) as an Early Biomarker and Therapeutic Target in Polycystic Ovary Syndrome

**DOI:** 10.3390/ijms21218191

**Published:** 2020-11-01

**Authors:** Xianqin Qu, Richard Donnelly

**Affiliations:** 1School of Life Sciences, University of Technology Sydney, Ultimo, NSW 2007, Australia; 2School of Medicine, University of Nottingham, Derby DE22 3DT, UK; richard.donnelly@nottingham.edu.uk

**Keywords:** adolescents, hepatic lipogenesis, human sex hormone-binding globulin, insulin resistance, non-alcoholic fatty liver disease, polycystic ovary syndrome

## Abstract

Human sex hormone-binding globulin (SHBG) is a glycoprotein produced by the liver that binds sex steroids with high affinity and specificity. Clinical observations and reports in the literature have suggested a negative correlation between circulating SHBG levels and markers of non-alcoholic fatty liver disease (NAFLD) and insulin resistance. Decreased SHBG levels increase the bioavailability of androgens, which in turn leads to progression of ovarian pathology, anovulation and the phenotypic characteristics of polycystic ovarian syndrome (PCOS). This review will use a case report to illustrate the inter-relationships between SHBG, NAFLD and PCOS. In particular, we will review the evidence that low hepatic SHBG production may be a key step in the pathogenesis of PCOS. Furthermore, there is emerging evidence that serum SHBG levels may be useful as a diagnostic biomarker and therapeutic target for managing women with PCOS.

## 1. Introduction

Polycystic ovary syndrome (PCOS) is a complex, common reproductive and endocrine disorder affecting up to 10% of reproductive-aged women [1]. Common symptoms of PCOS include irregular menstrual cycles (oligo-ovulation), hyperandrogenism, hirsutism with acne vulgaris and polycystic ovaries [2]. In addition, women with PCOS are often affected by metabolic abnormalities, including obesity, insulin resistance, dyslipidemia, systemic inflammation, non-alcoholic fatty liver disease (NAFLD), and hypertension, as well as an increased risk for type 2 diabetes and cardiovascular disease [3].

The complex and heterogeneous characters of PCOS give challenges in making the diagnosis. Various diagnostic criteria for PCOS and its associated phenotypes have been published. Utilizing the most widely accepted criteria [4,5,6,7,8], PCOS can be described in terms of four different phenotypes (Figure 1). The phenotype which includes hyperandrogenism, menstrual irregularities/anovulation and polycystic ovaries (HA + M + PCO) is considered to be the most severe form of PCOS which significantly increases the risk for infertility and adverse pregnancy outcomes [9].

While the aetiology of PCOS remains unclear, it is likely to be multifactorial. No single cause has been identified; however, multiple lines of evidence suggest a complex interplay between genetic and environmental factors [10,11]. Currently, there is no single treatment for PCOS and in clinical practice the therapeutic strategy is usually focused on individualised treatment to manage clinical symptoms and reduce metabolic and cardiovascular risk [12].

Women with PCOS may present with a number of reproductive, metabolic, psychological and anthropometric complications [13,14]. For example, ovarian dysfunction leads to anovulatory infertility. Furthermore, for those women who do become pregnant there is an increased risk of miscarriage, foetal complications and gestational diabetes [9,15,16,17].

Epidemiological studies indicate that the majority of PCOS begins during pubertal growth [18,19]. To prevent PCOS associated comorbidities and complications, it is important to develop an approach that targets the mechanisms behind PCOS at its earliest stages, e.g., in adolescents.

## 2. PCOS Diagnosis

### 2.1. A Case Report

A 19-year-old girl presents with a history of irregular menses for 4 years. She had undergone puberty that was normal in both timing and development, with menarche at 12 years of age. At 16 years of age, she started irregular menses with the cycle length varying between 20 and 60 days and bleeding for about 6 days. She reports feeling depressed about facial acne. Her weight is 80 kg and height 1.75 m; body-mass index is 26.2. Physical examination confirms no hirsutism but she does have severe facial acne. 

Investigations show that her follicle-stimulating hormone (FSH) and luteinizing hormone (LH) levels are in the normal ranges with low levels of estradiol (OEST2) and progesterone. Total testosterone is 1.1 nmol/L (NR 0.6–2.5), dehydroepiandrosterone sulfate (DHEAS) 5.1 umol/L (NR 3.3–12), and androstenedione 7.7 nmol/L (NR 4.9–14). The calculated free testosterone 33.6 pmol/L (NR 3.5–46.0) level is in the normal range but the serum SHBG concentration is only 8 nmol/L (NR 20-118). Her fasting plasma glucose is 4.9 mmol/L, HbA1c 5.1%, and fasting insulin level is raised at 35 mU/L (NR < 10). The lipid profile is normal but liver function tests are raised: serum ALT 58 U/L (NR < 30) and abdominal ultrasound shows a pattern of diffuse fatty infiltration of the liver.

In addition, the ovarian morphology was unremarkable on ultrasound but an accurate antral follicle count could not be defined (a transvaginal scan was declined by the patient).

The diagnosis of PCOS in this woman is supported by the history of irregular menses with a raised BMI and evidence of acne vulgaris. Facial acne is often a complication of androgen excess but her androgen levels were within the normal range. However, her SHBG level was very low, thereby increasing androgen activity which may adversely affect her skin. Investigations also revealed this young woman had evidence of NAFLD and insulin resistance (as shown indirectly by the compensatory hyperinsulinaemia).

The history of irregular menstruation would indicate ovarian dysfunction but in this young woman there was neither biochemical hyperandrogenism nor polycystic ovary morphology (PCOM) on ultrasound at this stage.

A recent clinical study recommended more simplified criteria for making a diagnosis of PCOS, based on testosterone as a single marker of hyperandrogenaemia alongside the key symptoms of oligomenorrhoea and hyperandrogenism as defined by either testosterone levels above a threshold and/or the presences of hirsutism [20].

Thus, a reasonable question raised by this case history is whether she should have a PCOS diagnosis? If yes, how do we prevent progression from early stages to the more classic and severe phenotype (HA + M + PCO)?

### 2.2. Clinical Features and Biochemical Assessments of PCOS Adolescents

Although PCOS often impacts fertility, the most common manifestations in teenagers include menstrual disturbances related to ovarian dysfunction and dermatological complaints such as hirsutism and acne. The criteria for PCOS diagnosis can be problematic when applied to younger women [18,21]. For example, the polycystic ovary is characterized by an excessive number of small antral follicles but this appearance is common in normal adolescents. Polycystic ovary morphology (PCOM) is best identified and evaluated using more invasive imaging techniques but this may not be acceptable in younger patients.

Thus, a diagnosis of PCOS in teenagers is often based on clinical manifestations and biochemical evidence of hyperandrogenemia. The diagnosis may be overlooked if total-testosterone and androstenedione levels are normal.

Manifestations of hyperandrogenemia are often related to low levels of SHBG. A meta-analysis of 16 independent studies showed that serum SHBG levels are decreased in young women with PCOS, especially obese adolescents [22]. Another study evaluated 106 young women with PCOS and found low levels of SHBG associated with low levels of HDL-Cholesterol but independent of obesity [23].

The progression of PCOS features from adolescence to early adulthood is poorly documented because of the paucity of cohort studies with long-term follow-up. One clinical study suggested that girls presenting with menstrual irregularity and acne at 16 years of age, with or without hyperandrogenism, were more likely to suffer from PCOS and infertility problems 10 years later than were non-symptomatic teenagers [24]. In addition, the endocrine and reproductive abnormalities place these young women at high risks for infertility, type 2 diabetes and cardiovascular disease. Thus, advances in understanding the pathogenesis of PCOS, e.g., the role of SHBG in the early development of this syndrome, may help clinicians to detect PCOS tendencies and intervene earlier to improve metabolic and reproductive outcomes.

## 3. Pathophysiological Factors that Regulate SHBG Production

### 3.1. The Biochemistry and Endocrinology of SHBG

SHBG is a 90–100 KDa homodimeric glycoprotein with two identical peptide chains that is encoded by a single gene on the short arm of chromosome 17 [25]. Circulating SHBG is produced primarily by hepatocytes, however, the gene is also expressed in the brain, uterus, breast, ovary, placenta, prostate and testis [26,27]. SHBG binds and transports testosterone, estradiol and other sex steroids in the plasma, reduces their metabolic clearance rate and affects their bioavailability [28]. SHBG exhibits high affinity for testosterone and a low affinity for estradiol. The relative binding affinity of various sex steroids for SHBG is dihydrotestosterone (DHT) > testosterone > androstenediol > estradiol > estrone [28].

The free active testosterone concentrations in plasma are very much influenced by SHBG concentrations because only 1–2% of testosterone in the circulation is free (unbound) and active; 65% is bound to SHBG and the rest is bound to albumin [28]. Therefore, women with low SHBG can have normal total testosterone levels but elevated bioavailable- and free-testosterone levels (see Table 1, adapted from [29]).

### 3.2. SHBG Expression and Production

SHBG is present in the fetal circulation and in cord blood where levels are similar in males and females [30]. Some cross-sectional studies indicate that SHBG levels rise substantially from birth to early childhood and then decline at puberty, more so in boys than in girls, probably due to androgens which are known to suppress SHBG levels [31].

SHBG levels in adulthood are higher in women than in men, which is probably due to estradiol since contraceptive therapy increases SHBG concentrations [32]. A longitudinal study of the premenopausal transition showed that SHBG significantly decreased in women during the menopause, which in turn correlates negatively with BMI and bone mineral density [33].

Because circulating SHBG is mainly secreted by the liver, better understanding of SHBG expression in the liver may open up new diagnostic and therapeutic approaches. Hepatocytes are the primary site for synthesis of SHBG [25]. Experimental studies using a human hepatoma cell line (HepG2) showed that SHBG expression is negatively affected by monosaccharides (glucose or fructose), insulin and androgens [34,35,36]. In contrast, thyroid and estrogenic hormones and the phytoestrogens increase SHBG production by up-regulating the expression of hepatocyte nuclear factor 4 alpha (HNF-4α) which plays a critical role in controlling the SHBG promoter activity [37,38]. HNF-4α is the most significant transcription factor that activates SHBG expression in the liver by binding to the cis-element DR1- and DR3-binding sites located upstream of the SHBG promoter [39]. The level of HNF-4α is inversely related to lipogenesis in the liver. Hyperinsulinemia inhibits HNF-4α expression and reduces the synthesis and production of SHBG in the liver [40].

### 3.3. SHBG and Insulin Resistance 

SHBG is not only a binding protein for testosterone, estradiol and other sex hormones, it also functions as a hormone or signal transduction factor itself. In vitro studies using cellular models of human insulin resistance demonstrated that decreased expression of SHBG protein and mRNA levels, along with decreased mRNA and protein levels of IRS-1, IRS-2, PI3Kp85a and GLUT-3 and GLUT-4, indicate that SHBG may down-regulate the PI3K/AKT pathway involved in the development of local and systemic insulin resistance [41].

An in vivo animal study using SHBG transgenic mice compared with wild type mice found that overexpression of SHBG protects against high fat diet (HFD) induced obesity and insulin resistance, as evidenced by lower glucose profiles during glucose tolerance tests and insulin tolerance tests [42]. Overexpression of SHBG also abrogated the increase in insulin, leptin and resistin levels, as well as the reduction in adiponectin induced by HFD [42].

Clinical investigations in both boys and girls during puberty have shown that SHBG is a strong predictor of insulin sensitivity and a decline in SHBG predicts the development of insulin resistance [43,44]. The Study of Women’s Health Across the Nation (SWAN) revealed that increased liver fat and decreased SHBG associated with increased metabolic risk in midlife, while there was also a negative association between SHBG and insulin among women with fatty livers [45].

SHBG regulates the bioactivity of sex hormones. Insulin resistance leads to hyperinsulinemia, which in turn causes reproductive dysfunction by down-regulating SHBG production [46]. Furthermore, therapies which reverse insulin resistance, for example, metformin, are associated with an increase in SHBG concentrations in women with PCOS [47]. Accordingly, SHBG has emerged as a biomarker of insulin resistance. However, the mechanism for this association remains unclear. In vitro studies using hepatoma cell lines demonstrate that high concentrations of insulin inhibit SHBG production [48]. Also, in vivo inhibition of insulin secretion with diazoxide leads to increased SHBG concentrations, suggesting that insulin inhibits hepatic synthesis of SHBG [49]. Taken together, these data imply that low SHBG may be a consequence rather than a cause of insulin resistance. An unsupervised phenotypic clustering analysis using data from PCOS genome-wide association study has identified two distinct PCOS subtypes: a “reproductive” group that presents higher SHBG levels with relatively low BMI and insulin levels, and a “metabolic” group that is characterized by higher BMI, glucose, and insulin levels with lower SHBG levels [50]. This finding highlights that PCOS is a heterogeneous reproductive disorder with different underlying biological mechanisms. The metabolic phenotype (non-reproductive group) with insulin resistance and low SHBG levels is at a high risk for infertility.

Insulin resistance is a condition in which the cellular responses to a given ambient insulin concentration are decreased relative to a normal control [51]. Insulin resistance has diverse manifestations in different tissues, mainly, skeletal muscle, liver, and white adipose tissue. Hepatic insulin resistance exacerbates hepatocellular effects on de novo lipogenesis (DNL) through transcriptional upregulation of several genes of DNL, including sterol regulatory element binding protein 1c (SREBP-1c), which promotes DNL by enhancing the transcription of several lipogenic enzymes, notably adenosine triphosphate citrate lyase (ACL), acetyl-CoA carboxylase 1 (ACC1), fatty acid synthase (FAS), and stearoyl-CoA desaturase (SCD1) [52].

The increased hepatic lipogenesis in insulin resistant states inhibits the expression of HNF-4α, thus down-regulating SHBG gene transcription and SHBG production [53]. A study using liver samples from nondiabetic obese patients with NAFLD demonstrated that triglycerides accumulate in the liver and insulin resistance (assessed by homeostatis model assessment, HOMA-IR) was inversely related to SHBG mRNA and to HNF4 α mRNA as well as to circulating SHBG levels [54]. Furthermore, these mRNAs, as well as serum SHBG levels, were negatively correlated with hepatic triglyceride content and ACC activity [54]. This may explain the molecular mechanism by which hyperinsulinemia downregulates HNF-4α expression, thereby reducing hepatic SHBG production by up-regulating ACC-lipogenesis.

Because insulin resistance is often accompanied by nutrient oversupply, high dietary intake of monosaccharides can induce low serum SHBG levels through increasing hepatic lipogenesis [55]. It has been shown that dietary fructose increases levels of the enzymes involved in DNL, even more strongly than HFD, and that fructose metabolism stimulating DNL is a central abnormality in NAFLD [55,56].

In addition, peroxisome proliferator-activated receptor-γ (PPAR-γ) competes with HNF-4α for a binding site (DR3) of the SHBG promoter [57], thus, PPAR-γ acts as an inhibitor of SHBG expression. SHBG production can thus be increased by lowering the activity of PPAR-γ [57]. However, in a clinical study of thiazolidinediones, PPAR-γ agonists, these drugs improved anovulation and hisutism associated with an increase in plasma SHBG levels in patients with PCOS [58]. This result however may be attributed to improved insulin resistance and metabolic abnormalities, implying insulin resistance is the causative factor for low SHBG.

Obesity and metabolic syndrome are associated with adipocyte insulin resistance which impairs suppression of lipolysis, leading to ectopic lipid deposition in non-adipose tissues and resultant “lipotoxicity” [59]. Adipose tissue insulin resistance increases lipolysis, which in turn releases non-esterified fatty acids (NEFAs) and lipid intermediates which fuel gluconeogenesis and lipogenesis, subsequently inhibiting HNF-4α and reducing SHBG production.

Excessive ectopic lipid accumulation also leads to local inflammation and worsening of insulin resistance. In the liver, visceral adipose depots trigger adipocyte-hepatocyte crosstalk in a bidirectional effect: inflammatory cytokines released from adipose fat, such as tumor necrosis factor-α (TNFα), can impair hepatic insulin signaling and promote accumulation of intrahepatic triglyceride, leading to inhibition of HNF-4α mRNA through activating nuclear factor-kB (NF-kB) [60]. In addition, the adipose inflammatory cytokine interleukin-1β (IL-1β) can inhibit HNF-4α expression by activating the MEK-1/2 and JNK MAPK pathways [61].

Adiponectin increases HNF-4α and SHBG levels through AMPK activation [62]. Lower SHBG levels are associated with lower adiponectin levels in patients with obesity and metabolic syndrome, possibly by increasing hepatic lipogenesis thereby leading to downregulation of HNF-4α [62,63]. Thus, adipose tissue insulin resistance associated with cellular inflammatory factors negatively affect hepatic production of SHBG and accelerate metabolic and reproductive disorders, such as NAFLD/NASH, PCOS and type 2 diabetes.

Conversely, in vitro studies using adipocytes and macrophages show that SHBG suppresses inflammation and lipid accumulation in macrophages and adipocytes, which might be among the mechanisms underlying the protective effect of SHBG, that is, its actions which reduce the incidence of metabolic syndrome and its complications [64]. Insulin resistance with various factors negatively regulating SHBG expression and production are summarized diagrammatically in Figure 2.

## 4. SHBG: An Important Biomarker in Metabolic and Reproductive Disorders

### 4.1. SHBG and Metabolic Syndrome

Interest in SHBG has increased in recent years because lower circulating levels of SHBG are inversely associated with obesity, insulin resistance (measured by HOMA-IR), metabolic syndrome, type 2 diabetes, and gestational diabetes [65,66,67]. In addition, low SHBG is associated with a greater coronary artery calcium score and increased risk for cardiovascular disease (CVD) in post-menopausal women [68,69]. Reductions in SHBG may be attenuated by low GI diets with low sugar and high fibre content among postmenopausal women [70]. The beneficial effects of SHBG on cardiovascular disease events in men is so far unproven [71], however, data on 2563 community-dwelling men (35 to 80 years) showed that elevated SHBG was associated with both a greater risk of CVD and an increased CVD mortality among all men and men > 65 years [72].

### 4.2. SHBG and NAFLD

NAFLD is characterized by two clinical entities: simple steatosis and non-alcoholic steatohepatitis (NASH), while simple steatosis accounts for 80–90% of cases of NAFLD [73]. Liver biopsy currently remains the gold standard for the diagnosis of NASH but numerous non-invasive biomarkers, including mild elevation of transaminases (AST, ALT and GGT) and/or ultrasound imaging are suggested as first-line screening tools for defining steatosis in a selected population, e.g., people with metabolic syndrome and PCOS [74,75,76].

Many lines of study have postulated that a reduction in SHBG levels in patients with metabolic syndrome plays a role in the development of NAFLD, which is effectively the hepatic manifestation of metabolic syndrome. Thus, SHBG is being viewed as a sensitive biomarker of NAFLD. Saéz-López et al., using SHBG-C57BL transgenic mice compared with wild control and ksJ-db/db mice, showed that SHBG could play a role in arresting the progression of NAFLD by downregulating key lipogenic enzymes in the liver, such as ACC and PPARγ, thus reducing lipogenesis [77,78]. These findings suggest that SHBG might play an important role in preventing the development of NAFLD and that SHBG levels could be used as a sensitive biomarker for identifying NAFLD.

In addition, enhancing SHBG expression may be a therapeutic strategy for the treatment of NAFLD. A recent longitudinal cohort study examined associations of circulating SHBG, estrogens and androgens with key histologic features of pediatric, biopsy-confirmed NAFLD and the results showed that lower SHBG levels were inversely associated with steatosis severity in boys and girls, and with portal inflammation in girls only [79], indicating that adipocyte-hepatocyte inflammatory crosstalk may exist in women with NAFLD.

### 4.3. SHBG Levels in Adult Women with PCOS

A recent systematic review and meta-analysis of 59 registered clinical trials indicated a positive correlation between metabolic syndrome and PCOS [80]. Interestingly, the meta-analysis found, in good and fair quality studies, that women with PCOS have had obesity related metabolic abnormalities associated with significantly lower SHBG levels but not with indices of hyperandrogenism [81], which highlights the possibility that decreased SHBG occurs prior to increased androgens in PCOS. Furthermore, therapeutic intervention with metformin, myo-inositol and D-chiro-inositol in women with PCOS increased serum levels of SHBG and were associated with improved ovarian function and metabolism (reduction of BMI, HOMA-IR and LDL-C) [47,82]. Therefore, serum SHBG levels may prove to be a useful biomarker for the diagnosis and treatment of PCOS [83].

### 4.4. SHBG Levels in PCOS Adolescents

With westernized lifestyles worldwide, obesity has greatly increased among teenagers and children. Obesity impacts the development of girls during puberty and increases the risk of PCOS [84]. Clinical observation has showed that childhood obesity is the initial sign of insulin resistance and a precursor of PCOS [85,86]. The relationship between obesity and PCOS is partly related to the negative influence of obesity on SHBG synthesis and secretion, which in turn increases testosterone bioavailability. The manifestations of PCOS often begin during adolescence [87], suggesting that puberty/adolescence is a critical developmental stage during which the pathophysiology of PCOS unfolds.

However, because the physiological changes of normal puberty overlap with the findings of PCOS, a diagnosis of PCOS in adolescents remains a controversial topic. Nonetheless, the degree of pre-pubertal obesity increases the risk of developing PCOS. A population-based study of adolescent girls (ages 15–19 years) suggested that the prevalence of PCOS (by NIH criteria) was approximately 3.85, 10.25, and 23.10% in overweight, moderately obese, and extremely obese girls, respectively [88]. Furthermore, the lower serum SHBG levels with metabolic syndrome have been identified as a risk of PCOS in adolescents [84,85,86]. Therefore, SHBG levels could represent a useful and practical test to screen young women for PCOS, in particular in adolescents.

## 5. Is SHBG a Link between NAFLD and PCOS?

### 5.1. The Role of SHBG and NAFLD in the Development of PCOS

Given the shared mechanisms underpinning the development of metabolic syndrome in both PCOS and NAFLD, epidemiological data indicate a higher prevalence of NAFLD in women with PCOS ranging from 34% to 70% compared with 14% to 34% in healthy women [89]. Thus, NAFLD can be considered as a hepatic complication of PCOS [90]. A retrospective longitudinal cohort study utilizing a large primary care database in the United Kingdom, evaluating NAFLD rates in 63,120 women with PCOS and 121,064 age-, BMI-, and location-matched control women found a significantly high prevalence of NAFLD in women with PCOS compared with the controls (*p* < 0.001) [91]. This large clinical study has pointed out that androgen excess contributes to the development of NAFLD in women with PCOS [91].

However, other clinical trials have shown that decreased serum concentrations of SHBG, but not testosterone, are associated with the metabolic syndrome in overweight and obese women with PCOS [81]. For example, a case-control study conducted in China showed that a high free androgen index (calculated by total testosterone × 100/SHBG) is associated with an increased risk of NAFLD, suggesting that low SHBG levels increase the risk of NAFLD in Chinese women with PCOS [92].

Other work has also reported that PCOS is more common among young women with biopsy-proven NAFLD and decreased plasma SHBG levels, regardless of overweight/obesity status [93] but insulin resistance and dyslipidaemia showed a reverse relationship with plasma levels of SHGB in women with PCOS [94].

Although there is frequent co-existence of PCOS and NAFLD, it remains unclear if NAFLD causes PCOS or if PCOS causes NAFLD or if the causative relationship is bidirectional.

Given the growing prevalence of NAFLD associated with PCOS in young women [95], it is important to clarify any causative relationship by further studying the molecular pathways by which hepatic fat contributes to decreased SHBG production.

### 5.2. Low SHBG Levels May Lead to Ovarian Dysfunction and PCOS

That low SHBG levels are often associated with fatty liver disease and hyperinsulineamia in adolescents with mild PCOS may reflect a novel hepato-ovarian axis [96]: reduction of hepatic SHBG synthesis in NAFLD causes increased androgen bioavailability as the initial stage of PCOS, but persistent increases in free androgen in the blood stream eventually causes ovarian dysfunction with excess secretion of androgens (hyperandrogenism). Thus, distinct states of “hepatic hyperandrogenism”, and “ovarian hyperandrogenism” may be described.

The mechanism by which hepatic hyperandrogenism links to ovarian hyperandrogenism may involve disrupting the negative feedback regulation in the hypothalamic–pituitary–ovarian axis [96]. Persistently increased circulating free androgen concentrations due to low SHBG can reduce the hypothalamus sensitivity to LH pulses, leading in turn to the excessive release of gonadotropin-releasing hormone which stimulates the pituitary gland to enable LH release.

Adolescents with mild PCOS present with increased GnRH and LH pulse frequency and amplitude, as well as an increased LH to FSH ratio (>3:1) but this is not part of the diagnostic criteria [97]. This increase is exaggerated in girls with a predisposition to PCOS, which further amplifies androgen production by ovarian theca cells, thus leading to ovarian hyperandrogenism. Reduced hepatic SHBG levels in NAFLD may trigger a cascade of excessive androgen production in adolescents, resulting in the development of PCOS.

In addition, liver dysfunction in NAFLD may affect sex steroid metabolism to worsen the hyperandrogenism. Fatty liver disease with decreased SHBG levels may play a causative role in the development of PCOS in adolescents.

Several studies have also demonstrated that insulin resistance may directly induce ovarian dysfunction, leading to hyperandrogenism and ovarian lesion in PCOS [98]. Hyperinsulinemia associated with obesity and increased proinflammatory cytokines can also lead to apoptosis of ovarian granulosa cells and occlusion of follicles [99], resulting in anovulation and infertility.

Experimental studies also show that proinflammatory cytokines are capable of stimulating proliferation of androgen producing theca cells to promote hyperandrogenism [100,101] through upregulating CYP17 (the ovarian steroidogenic enzyme responsible for androgen production) which can be normalized by naturally occurring anti-inflammatory agents [102,103].

It can be postulated that increased hepatic lipogenesis inhibits hepatic SHBG synthesis, which in turn leads to relative and absolute hyperandrogenemia and ovarian dysfunction in PCOS (Figure 3). Conversely, a low carbohydrate diet, insulin sensitizer, and phytoestrogens can all enhance SHBG production through reduction of hepatic lipogenesis and improving fatty liver to prevent progression of ovarian pathology and restore reproductive function [47,70,102,103,104,105].

### 5.3. SHBG: An Emerging Biomarker with Potential Utility in Detection, Surveillance and Treatment of PCOS

PCOS is complex with reproductive and metabolic features and ~75% of adult women with PCOS present with infertility [106]. A prospective cohort study suggests that women with PCOS, particularly the phenotype with HA + M + PCO, have a significantly increased risk for adverse pregnancy and neonatal outcomes including miscarriages, gestational diabetes, pre-eclampsia, and reduced weight babies [9].

The majority of PCOS pathology begins in adolescence. A clinical study suggests that girls presenting with menstrual irregularity and acne, with or without hyperandrogenism at 16 years, were more likely to suffer from PCOS and fertility problems at age 26 years than the non-symptomatic girls [24]. Therefore, therapeutic interventions at the earliest stage of onset (e.g., at age 16) may reverse the pathology of the disease and prevent fertility problems 10 years later. However, a diagnosis of PCOS based on the Rotterdam criteria often does not apply among younger women [21].

In clinical practice, SHBG is a sex steroid. Because the bioactivity of androgens is determined by the free-testosterone, SHBG levels are important in the evaluation of hyperandrogenism. However, SHBG is not yet an established component of PCOS diagnostic criteria. As in the case described, a low plasma SHBG is often the only abnormal parameter of the androgen panel tests in teenage girls with menstrual irregularity and acne; SHBG reduction may be an early manifestation of PCOS, prior to an overt phenotype emerging. One the other hand, the role of SHBG in the evaluation of adult PCOS may be less important because adult women with or without PCOS often take the combined oral contraceptives (COCs) either for pregnancy prevention or for menstrual regularity and hyperandrogenism.

COCs have been used for many years as a first line treatment to reduce the manifestations of androgen excess and regulating menstrual cycles in women with PCOS [106]. It has been reported that COC containing of 30 μg ethynilestradiol and 3 mg drospirenone for six months significantly increased plasma SHBG from 37.31 nmol/L to 179.01 nmol/L, with an average increase of 141.7 nmol/L (*p* = 0.002) but COC use has also been associated with impaired fasting glucose, insulin resistance and increased risk of thromboembolism disease [107,108].

## 6. Conclusions

This review has discussed the inter-relationships of SHBG, NAFLD and PCOS and the associated molecular mechanisms. A low SHBG level may be only abnormal biochemical test in teenage girls presenting with menstrual irregularity and acne, who may later develop one or other of the diagnostic phenotypes of PCOS. More clinical studies are needed to assess the usefulness of SHBG as a biomarker for identifying young women who may later develop PCOS.

## Figures and Tables

**Figure 1 ijms-21-08191-f001:**
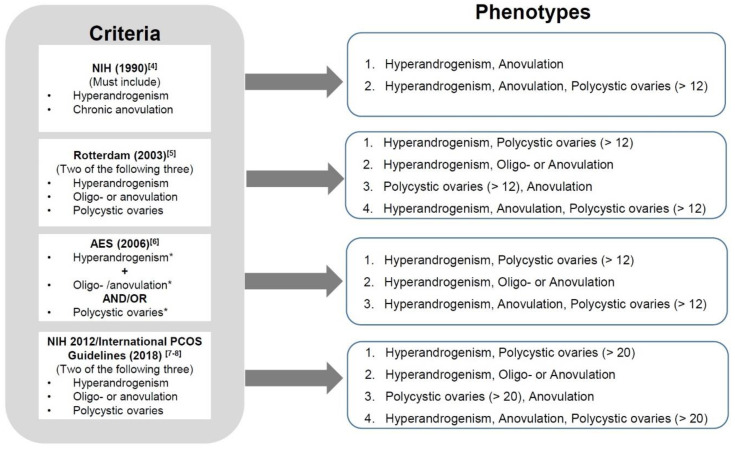
Diagnostic criteria and 4 main phenotypes of polycystic ovary syndrome (PCOS). *Hyperandrogenism: either testosterone above threshold and/or the presences of hirsutism or acne or androgenic alopecia. *Oligo-anovulation: eight or less ovulations per year. *Polycystic ovaries: ovarian volume > 10 mL and/or antral follicles number as defined above and less than 9 mm in size at least in one ovary.

**Figure 2 ijms-21-08191-f002:**
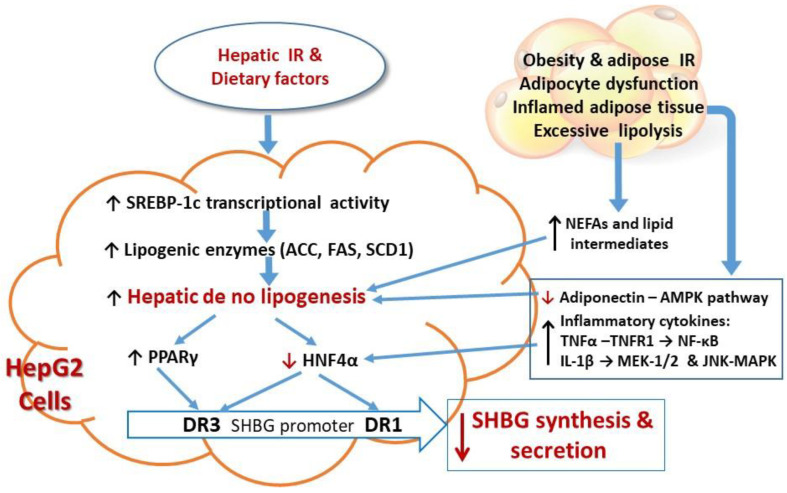
The molecular mechanism by which insulin resistance mediates various factors/pathways that reduce SHBG production, in part by down-regulating HNF-4α in the liver. (1) Hepatic insulin resistance with dietary factors (fructose and sucrose) increases monosaccharide delivery to the liver to cause reduction of hepatic HNF-4α levels and SHBG production through stimulating sterol regulatory element binding protein 1c (SREBP-1c), which promotes hepatic de no lipogenesis (DNL) by enhancing the transcription of several lipogenic enzymes, notably adenosine triphosphate citrate lyase (ACL), acetyl-CoA carboxylase 1 (ACC1), fatty acid synthase (FAS), and stearoyl-CoA desaturase (SCD1). (2) Obesity and insulin resistance increase adipose lipolysis to deliver NEFAs and lipid intermediates to promote gluconeogenesis and lipogenesis, subsequently inhibiting HNF-4α and reducing SHB. (3) Obesity with adipose insulin resistance increases pre-inflammatory cytokines: TNF-α downregulates HNF-4α expression through NF-κB activation; IL-1β decreases HNF-4α expression through the MEK-1/2 and JNK-MAPK pathways; adiponectin increases HNF-4α levels through AMPK activation. Reduction of adiponectin increases hepatic lipogenesis and thereby leading to downregulation of HNF-4α. (4) HNF-4α increases SHBG transcriptional activity by binding to the cis-elements DR1 and DR3 in the SHBG promoter. (5) Enhanced DNL inhibits HNF-4α and stimulates PPAR-γ expression which competes with HNF-4α for binding to DR3 and inhibits SHBG transcriptional activity, attenuating SHBG expression and reduced SHBG secretion. Words or arrows in red color are used to emphasize the key factors involved in insulin resistance mediated SHBG production.

**Figure 3 ijms-21-08191-f003:**
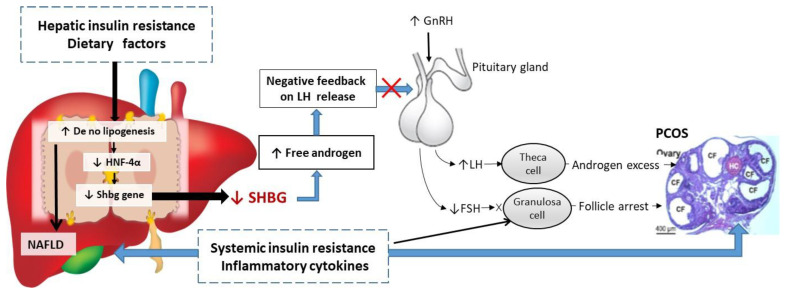
Schematic representation of the putative mechanisms linking low SHBG with the development NAFLD and PCOS. Increased hepatic *de no* lipogenesis inhibits SHBG synthesis through downregulating HNF-4α, leading to increasing androgen bioavailability and disrupting negative feedback relationship of hypothalamic–pituitary–ovarian axis, subsequently resulting in higher levels of LH and lower levels of FSH in most women with PCOS. Higher LH levels seem to cause hyperandrogenemia by exuberating androgens secretion from follicular theca cells, whereas lower FSH levels lead to anovulation. System insulin resistance associated with inflammatory cytokines directly induces granulosa cells apoptosis, leading to arrested follicular maturation, contribute to anovulation and multiple cystic follicles (CF) formation in PCOS.

**Table 1 ijms-21-08191-t001:** Effects of sex hormone binding globulin concentration on bioavailable and free testosterone.

	Patient 1	Patient 2	Reference Range
Total testosterone	50.4	48.9	11–56 ng/dL
SHBG	25.1	186.0	30–135 nmol/L
Bioavailable testosterone	26.0	5.8	4.1–22.6 ng/dL
Free testosterone	10.2	2.3	1.3–9.2 ng/mL

Patient 1 is a woman with insulin resistance; patient 2 is on treatment with oral contraceptives. Even though both patients have similar and normal total testosterone concentrations, patient 1 has higher bioavailable and free testosterone, while patient 2 has normal bioavailable and free testosterone.

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
