# Peer review of "Sex Hormone-Binding Globulin (SHBG) as an Early Biomarker and Therapeutic Target in Polycystic Ovary Syndrome"

_ijms, 2020, doi:10.3390/ijms21218191_

Round 1

Reviewer 1 Report

Data of Dapas et al., PLOS Medicine, 2020 should be cited and discussed. Low SHBG is connected to metabolic profile of PCOS patients. However, reproductive group of PCOS patients represent high SHBG concentration. The paper is worth publishing.

Author Response

Thanks to reviewer's comment and suggestion. A paragraph with reference #50 (DapasI et al. Distinct subtypes of polycystic ovary syndrome with novel genetic associations: An unsupervised, phenotypic clustering analysis. PLoS Med. 2020, 23, 17, e1003132) as below has been inserted in the text of Section 3.2, line 177-184.

“An unsupervised phenotypic clustering analysis using data from PCOS genome-wide association study has identified two distinct PCOS subtypes: a “reproductive” group that presents higher SHBG levels with relatively low BMI and insulin levels, and a “metabolic” group that is characterized by higher BMI, glucose, and insulin levels with lower SHBG levels [50]. This finding highlights that PCOS is a heterogeneous reproductive disorder with different underlying biological mechanisms and that the metabolic phenotype (non-reproductive group) with insulin resistance and low SHBG levels is at high risk for infertility.”

Reviewer 2 Report

The authors the present manuscript have reviewed the evidence that low hepatic SHBG production may be a key step in the pathogenesis of PCOS and that serum SHBG levels may be useful as a diagnostic biomarker and therapeutic target for managing women with PCOS. The manuscript is well written and contains enough review literatures that support the idea of using serum SHBG as a biomarker for PCOS in women. However, very recently similar reviews have been published that have discussed the inter-relationships of SHBG and PCOS. One of review by Jing-Ling Zhu from 2019 has also discussed the interaction between SHBG and the various complications of PCOS as well as the regulatory effect of HNF-4α on SHBG expression which has not been mentioned in the present review.

Author Response

Thanks for reviewer 2’s comment. We have read and leant knowledge from review article published by Zhu et al (Sex hormone-binding globulin and polycystic ovary syndrome. Clin Chim Acta, 2019, 499:142-148). In this review, we have endeavoured to collect data/ and summarize information from updated original studies. Zhu’s review article has been cited in this revised version, Section 3.2 of line 147-148, reference #39.  

"HNF-4α is the most significant transcription factor that activates SHBG expression in the liver by binding to the cis-element DR1- and DR3-binding sites located upstream of the SHBG promoter [39]".

Round 2

Reviewer 2 Report

The authors have now mentioned the last year's reference, but it would be better to distinguish the novelty of this review from the earlier review.

Author Response

The major novelty of this review compared to Zhu's publication is that using a real clinical case illustrates Sex hormone-binding globulin (SHBG) as an Early Biomarker And Therapeutic Target In Polycystic Ovary Syndrome in adolescents.

We do not believe that this novelty comparison should be included in revised manuscript. Therefore, the revised version of our manuscript remains no change.